# Outcomes in acute pulmonary embolism and their association with adherence to international recommendations around COVID-19 pandemic-induced hospital-strain: The experience in a Mexican tertiary care center

Juan José Rodríguez-Crespo[1], Eduardo Gutiérrez-León[2], Pedro Dammann-Beltrán[3], Jurhiat Alejandro Seaman-Gómez[4], Sergio Contreras-Garduño[5], Erick Yasar Zúñiga-González[6], José Guillermo Domínguez-Cherit[7], Thierry Hernández-Gilsoul[1], José de Jesús Vidal-Mayo[1], Fausto Alfredo Ríos Olais[5], Raúl Rivera Moscoso[8], Jorge Oseguera Moguel[9], Mónica Chapa Ibargüengoitia[10], Heber Rodríguez Bautista[7], Adrián Soto-Mota[3,11]*, José Sifuentes-Osornio[12]

1 Emergency Department and Continuous Institutional Attention, Instituto Nacional de Ciencias Médicas y Nutrición Salvador Zubirán, Mexico City, Mexico, 2 Department of Medical Education, Instituto Nacional de Ciencias Médicas y Nutrición Salvador Zubirán, Mexico City, Mexico, 3 Department of Neurology and Psychiatry, Instituto Nacional de Ciencias Médicas y Nutrición Salvador Zubirán, Mexico City, Mexico, 4 Metabolic Diseases Research Unit, Instituto Nacional de Ciencias Médicas y Nutrición Salvador Zubirán, Mexico City, Mexico, 5 Hematology and Oncology Department, Instituto Nacional de Ciencias Médicas y Nutrición Salvador Zubirán, Mexico City, Mexico, 6 Department of Nephrology and Mineral Metabolism, Instituto Nacional de Ciencias Médicas y Nutrición Salvador Zubirán, Mexico City, Mexico, 7 Tecnológico de Monterrey, Escuela de Medicina y Ciencias de la Salud; Instituto Nacional de Ciencias medicas y Nutricion Salvador Zubiran, Mexico City, Mexico, 8 Direction of Medicine, Instituto Nacional de Ciencias Médicas y Nutrición Salvador Zubirán, Mexico City, Mexico, 9 Department of Cardiology, Instituto Nacional de Ciencias Médicas y Nutrición Salvador Zubirán, Mexico City, Mexico, 10 Department of Radiology and Molecular Imaging, Instituto Nacional de Ciencias Médicas y Nutrición Salvador Zubirán, Mexico City, Mexico, 11 Tecnológico de Monterrey, School of Medicine, Mexico City, Mexico, 12 General Direction, Instituto Nacional de Ciencias Médicas y Nutrición Salvador Zubirán, Mexico City, Mexico

* adrian.sotom@incmnsz.mx

## Abstract

### Background

Lack of adherence to international recommendations leads to worse outcomes. During the COVID-19 pandemic, the total number and proportion of hospitalized patients increased, consequently straining hospital care and hindering adherence.

### Aims

To evaluate adherence to clinical guidelines at our center, before, during, and shortly after the COVID-19 pandemic-induced hospital strain, and its association with clinical outcomes, using a random, balanced retrospective cohort.

**Data availability statement:** There are ethical or legal restrictions on sharing de-identified data imposed by both Mexican law and our institutional IRB because, even if anonymised, clinical data is unique enough to have an inherent high risk of becoming identifiable. However, anonymised data for research purposes is available upon request to the corresponding author (adrian.sotom@incmnsz.mx). Since our study involved legally protected electronic medical records, only the requested and properly justified variables will be shared to ensure anonymity is preserved. To contact the IRB in charge of approving any research involving our institution's electronic medical records, contact: comite.etica.investigacion@incmnsz.mx.

**Funding:** The author(s) received no specific funding for this work.

**Competing interests:** The authors have declared that no competing interests exist.

**Abbreviations:** AIDS: Acquired Immunodeficiency Syndrome; COVID-19: Coronavirus Disease of 2019; CTPA: Computed Tomography Pulmonary Angiogram; PE: Pulmonary Embolism; PERT: Pulmonary Embolism Response Team; PESI: Pulmonary Embolism Severity Index; VTE: Venous Thromboembolism.

## Design

A balanced and randomized sample of 50 patients per year between 2019 and 2022 was drawn from electronic medical records and analyzed with multivariable and logistic regression models. The primary outcome was a composite of in-hospital death, hemodynamic decompensation within the first 7 days, and in-hospital bleeding.

## Results

The global non-adherence in our study was 45.4%. The main risk factors for non-adherence were any mortality risk classification above low-risk PE and a PESI class different from class I, with ORs of 3.47 (95% CI 2.07–5.82) and 1.57 (95% CI 1.04–2.37), respectively. In both periods (COVID-19 season and non-COVID-19 season), non-adherent management strongly correlated with the composite outcome, OR = 2.36 (95% CI, 1.23–4.54). Non-adherence was also associated with worse in-hospital outcomes, with an incidence rate of the composite outcome of 21.08 per 1000 days/person (95% CI 10.97–40.51) and 47.7 per 1000 days/person (95% CI 33.39–67.50), and an attributable risk of 1.09% (95% CI −8.47%−10.64%).

## Conclusions

Overall, our findings highlight the need to prioritize human and material resources to ensure adherence to the standards of care for PE patients.

## Introduction

Pulmonary embolism (PE) is the most severe form in the venous thromboembolism (VTE) spectrum representing a leading cause of cardiovascular mortality, just behind of ischemic heart disease (IHD) and stroke [1], and is the main cause of preventable death among hospitalized patients [2].

The annual incidence rates for PE range from 39 to 115 per 100 000 population [3], and have been steadily increasing over the past 2 decades, at least in high-income countries. An improvement in life expectancy, particularly among patients with conditions predisposing to venous thromboembolism (VTE), such as cancer, chronic obstructive pulmonary disease, and autoimmune diseases, may partly explain this finding. Other factors explaining this trend include the broader adoption of validated diagnostic algorithms, greater awareness, a lower threshold of clinical suspicion, and the standard use of computed tomographic pulmonary angiography (CTPA) [4].

Nonetheless, VTE has an age-standardized mortality rate ranging from 0 to 24 deaths per 100 000 population- years [3]. The total number of VTE-related deaths is estimated at 543,454 across the US annually, which is more than double the number of combined deaths due to AIDS (5,860), breast cancer (86,831), prostate cancer (63,636), and transport accidents (53,599) [5]. Simultaneously, VTE is associated

with high health-care costs, is the leading cause of disability-adjusted life-years in low- income and middle-income countries, and the second most common cause in high-income countries [6].

However, both, PE-related in-hospital death rate and age- standardized mortality from PE have been decreasing or plateauing over the past years, possibly reflecting the greater proportion of "low-risk" cases being diagnosed and an improvement in PE management. In contrast, PE associated with hemodynamic instability still portends unacceptably high rates of in-hospital or early death [4].

Despite the publication of evidence-based clinical practice guidelines for VTE, clinicians frequently encounter VTE scenarios for which data are sparse and optimal management is unclear. In particular, the optimal use of advanced therapies for acute VTE, including thrombolysis and catheter-based therapies, remains uncertain [7].

To date, adherence to guidelines often remains low, causing omission of recommended therapies, or providing additional interventions that might not necessarily help but can contribute to preventable harm, sub optimal patient outcomes or experiences, and waste of resources. Moreover, acceptance of and compliance with guidelines might be associated with better prognosis for patients with acute PE [8].

While adherence to clinical guidelines for the management of PE before, during and shortly after the COVID-19 pandemic-induced hospital strain has not been described yet, it has been documented that during the COVID-19 pandemic hospital strain had a negative impact in patient's outcomes possibly (and at least, partially) via hindering adherence to established guidelines [9].

Hereby, using a random and balanced retrospective cohort, we aimed to evaluate how adherence to clinical guidelines, before, during and shortly after the COVID-19 pandemic impacted on short-term clinical outcomes in PE patients.

## Methods

### Study design and setting

We conducted a retrospective observational cohort study in patients with the diagnosis of pulmonary embolism admitted to a tertiary care center in Mexico City. This study was approved on September 26, 2024, by Internal Review Ethics Board of the National institute of Medical Sciences and Nutrition Salvador Zubirán with the code 3944. Written consent was waived due to the retrospective nature of our study.

### Data collection

We collected demographics, clinical variables, and PE-related outcomes from the electronic medical records until in-hospital death or discharge from January 2019 to December 2022.

### Eligibility criteria

We included patients 18 years old or older admitted due to PE confirmed with a computed tomography pulmonary angiogram (CTPA) [10]. We excluded patients with chronic pulmonary embolism and patients with incomplete records.

### Exposure, definitions, and outcomes

Patients were stratified based on their year of admission (to draw the random samples), and then, they were sub-stratified depending on admission during the COVID-19 season or not. We defined COVID-19 season as the weeks in which hospital occupancy in Mexico City due to COVID-19 represented $\geq$ 70% according to data from the Severe Acute Respiratory Infection Network Information System of Mexico's Ministry of Health [11].

We registered baseline characteristics, patient hospital location, and severity (short term mortality risk) of acute PE at diagnosis. Short term mortality risk was classified into low risk, intermediate-low risk, intermediate-high risk, and high risk in compliance with the 2019 ESC Guidelines for the Diagnosis and Management of Acute Pulmonary Embolism [3]. In

addition to mortality risk, the PESI score was calculated by the investigators. We defined "Advanced treatments" as thrombolysis with any dose, catheter-based interventions (thrombolysis and/or clot fragmentation) and surgical embolectomy.

Although there is no consensus for the management of patients with acute pulmonary embolism specially in patients with a more severe disease, the PERT Consortium made recommendations for the Diagnosis, Treatment and Follow Up of Acute Pulmonary Embolism [12]. In this study, we defined adherence to international recommendations based on the compliance of the following seven items: 1) Use of systemic thrombolysis in patients with high-risk PE without contraindications to systemic thrombolysis. 2) Use of any advanced treatment (other than systemic thrombolysis) in patients with high-risk PE with any relative or absolute contraindication to systemic thrombolysis. 3) Use of any advanced treatment in patients with intermediate-high risk PE with evidence of possible further deterioration (hemodynamic stability with 1 or more of the following: altered Glasgow Coma Scale, abnormal vital signs, elevated lactate) but who did not meet the definition of high-risk PE, with low risk of bleeding prior to hemodynamic deterioration. 4) Use of advanced treatment in patients with intermediate-high risk PE following hemodynamic deterioration. 5) Use of UFH in intravenous infusion in patients with kidney dysfunction (GFR < 30 ml/min) and/or severe obesity (>120 kg) and/or high-risk or intermediate-high risk PE with subsequent use of some advanced treatment. 6) Use of LMWH in patients without kidney dysfunction (GFR < 30 ml/min) nor severe obesity (>120 kg) nor in high-risk or intermediate-high risk PE with subsequent use of some advanced treatment. And if obesity and/or kidney dysfunction were present if it was used with concomitant measurement of anti-Xa factor activity. 7) Placement of inferior vena cava filter in patients with any contraindication for anticoagulation within the first 3 days of diagnosis ([12–14]; S1 Fig).

The primary outcome was a composition of in-hospital complications (mortality, hemodynamic decompensation within 7 days of diagnosis, and major bleeding). The secondary outcome was to evaluate patterns of adherence to international recommendations around the COVID-19 season adjusted to the severity of acute PE. Data were accessed for research purposes between 01/10/2024 and 01/09/2025.

## Statistical analysis

The description of the baseline characteristics was reported using measures of central tendency and dispersion, and with frequency distribution measures for non-continuous variables. The comparison of the baseline characteristics between individuals in the COVID-19 and non-COVID-19 seasons was carried out with a Chi-squared test for dichotomous and nominal variables, while an ANOVA test was performed for continuous variables. If they did not meet the assumptions of the statistical tests, Fisher's exact test and the median extension were performed.

The estimation of proportions was carried out for each of the categories of non-adherence, considering both the COVID-19 and non-COVID-19 periods. Additionally, a multivariate model (including age, sex, year, COVID-19 period, hospital location, Charlson index, presence of any chronic-degenerative comorbidity, and adherence) was used to estimate the association of adherence with the composite outcome; The odds ratios and their 95% confidence intervals (95% CI) were calculated.

Subsequently, the frequency of adherence and non-adherence was estimated by categories of severity of PE episodes (low, intermediate-low, intermediate-high, and high) and by COVID-19 season as a Boolean variable.

Additionally, incidence rates according to COVID-19 season status for the composite outcome, and individually, for each of the components were estimated. The incidence rates for the composite outcome according to the severity categories of PE episodes and PESI were also estimated. The number of events per 1000 person-days, with their respective 95%CI, were calculated as well.

Finally, the rate ratio (RR) between adherence and non-adherence for the COVID-19 and non-COVID-19 season status was calculated, and their 95% CI was reported. The RR was also within the COVID-19 and non-COVID-19 season with adherence. From the latter, the population attributable risk of not adherence to international recommendations for the medical management of PE on the composite and individual outcome; through a logistic regression model adjusted for age and sex.

## Results

The baseline characteristics are outlined in Table 1. Most of our population had a medical problem, 31.% (n = 63) were obese, 25% (n = 51) had diabetes, 20.5% had active cancer, and 38.5% (n = 77) had high blood pressure. Almost one third of the patients had concomitant DVT, and 14.5% (n = 29) have had a previous VTE episode. Most patients were classified as a low or intermediate-low mortality risk, and they had a mean PESI score of 113.02 ± 33.84.

The population in the non-COVID-19 season was younger compared to the COVID-19 season population (median age; 54.5 VS 64.0 respectively, p = 0.01). Likewise, 57.0% were women in the non-COVID-19 season, while 37.7% were in the COVID-19 season (p = 0.007).

Overall, 63 (31.5%) were obese, 77 (38.5%) had hypertension, 51 (25.5%) lived with diabetes, and 178 (89.0%) had at least one identifiable risk factor. Furthermore, a higher proportion of previous thromboembolic events (22.1% VS 8.8%), deep vein thrombosis (39.5% VS 15.8%), and active cancer (31.4% VS 12.3%) were observed in the non-COVID-19 season.

Additionally, 12.8% and 9.7% of the PE events were unprovoked in the non-COVID-19 and COVID-19 seasons, respectively. In the laboratory and CTPA findings, it was noteworthy that 24 (60%) and 19 (47.5%) of those with hemodynamic decompensation had elevated high-sensitivity troponin I levels (>16 pg/mL) and evidence of right ventricular failure on CTPA, respectively.

When estimating the risk of mortality from PE, 50.0% and 23.3% had a low and intermediate-low risk in the non-COVID-19 season, while in the COVID-19 season it was 22.8% and 47.4%, respectively; and for the intermediate-high and high risk, it was 17.4% VS 10.2%, and 9.3% VS 9.6%, in the given order. Notably, The COVID-19 season had a higher proportion of hemodynamic decompensation (15.1% VS 23.7%), death (25.6% VS 36.8%) and non-adherence to PE guidelines (43.9% VS 46.3%, p = 0.76) (Table 1).

In the total study population, the highest overall non-adherence was observed for: 1) use of any advanced treatment in patients with high-risk PE with any contraindication to systemic thrombolysis, 2) placement of inferior vena cava filters, and 3) use of any advanced treatment in patients with intermediate-high risk PE with further risk of hemodynamic decompensation and with a low bleeding risk (S1 Table).

When evaluating the variables associated with the composite outcome in the multivariate model, only non-adherence showed a significant association with an odds ratio of 2.36 (95% CI, 1.23–4.54) (S2 Table).

Stratifying by the risk of mortality from PE and adherence to PE guidelines, an inversely proportional relationship was observed for both periods. In the low-risk category, 75.0% and 83.3% showed adherence to the guidelines for the non-COVID-19 and COVID-19 season, respectively. Contrastingly, in the high-risk category, very low adherence was observed (12.5% non-COVID-19 VS 0.0% COVID-19).

Likewise, when evaluating the number of adherence deviations to guidelines with respect to the mortality risk; a lower number of adherence deviations were documented in the low-risk category compared to the high-risk category (1 deviation, 90.4% VS 5.3%; 2 deviations, 9.6% VS 73.7%; 3 deviations, 0.0% VS 21.1%) (Fig 1).

Regarding the incidence rates per 1000 person-days of the composite outcome, similar values were observed as well as for individual outcomes when comparing non-adherence with adherence during the non-COVID-19 season. However, when comparing non-adherence and adherence during the COVID-19 season, higher incidence rates were calculated for both the composite and individual outcomes. This was observed as well when stratifying the incidence rate (1000 days/person) for the composite outcome by mortality risk for PE (Table 2).

There was no difference when comparing the rate ratios (RR) of the incidences of the composite outcome (1.10 CI95% [0.39–3.01]), hemodynamic decompensation at day 7 (1.22 [0.28–5.30]) and in-hospital death (1.22 [0.43–3.47]) between adherence and non-adherence status during the non-COVID-19 season. However, during the COVID-19 season, non-adherence implied a two-fold increase in the incidence for the same outcomes. This trend on the composite outcome was maintained after stratifying by mortality risk.

**Table 1. Baseline characteristics, by groups\*.**

| Characteristics | Overall N = 200 | Non COVID-19 season (N = 86) | COVID-19 season (N = 114) | P |
|---|---|---|---|---|
| Age (yr), median (interquartile range) | 60 (46-72) | 54.50 (31-66) | 64.00 (53-75) | 0.001 |
| Men, *N* (%) | 108 (54.00) | 37 (43.02) | 71 (62.28) | 0.007 |
| Patient type, *N* (%) | | | | 0.08 |
| Medical | 175 (87.50) | 70 (81.40) | 105 (92.11) | |
| Surgical | 11 (5.50) | 7 (8.14) | 4 (3.51) | |
| Both | 14 (7.00) | 9 (10.47) | 5 (4.39) | |
| Risk factors, *N* (%) | | | | |
| Obesity | 63 (31.50) | 22 (25.58) | 41 (35.96) | 0.12 |
| Diabetes | 51 (25.50) | 19 (22.09) | 32 (28.07) | 0.34 |
| COPD | 6 (3.00) | 4 (4.65) | 2 (1.75) | 0.23 |
| Hypertension | 77 (38.50) | 29 (33.72) | 48 (42.11) | 0.23 |
| Chronic Kidney Disease | 17 (8.50) | 7 (8.14) | 10 (8.77) | 0.87 |
| Recent Major Surgery | 13 (6.50) | 9 (10.47) | 4 (3.51) | 0.05 |
| Chronic Heart Failure | 10 (5.00) | 5 (5.81) | 5 (4.39) | 0.65 |
| Immobilization | 27 (13.50) | 15 (17.44) | 12 (10.53) | 0.16 |
| Previous VTE | 29 (14.50) | 19 (22.09) | 10 (8.77) | 0.008 |
| Concomitant DVT | 52 (26.00) | 34 (39.53) | 18 (15.79) | <0.001 |
| Active Cancer | 41 (20.50) | 27 (31.40) | 14 (12.28) | 0.001 |
| Mortality risk, *N* (%) | | | | <0.001 |
| Low | 69 (34.50) | 43 (50.00) | 26 (22.81) | |
| Intermediate-low | 74 (37.00) | 20 (23.26) | 54 (47.37) | |
| Intermediate-high | 38 (19.00) | 15 (17.44) | 23 (20.18) | |
| High | 19 (9.50) | 8 (9.30) | 11 (9.65) | |
| PESI, *N* (%) | | | | <0.001 |
| I | 10 (5.00) | 8 (9.30) | 2 (1.75) | |
| II | 29 (14.50) | 20 (23.26) | 9 (7.89) | |
| III | 46 (23.00) | 22 (25.58) | 24 (21.05) | |
| IV | 49 (24.50) | 10 (11.63) | 39 (34.21) | |
| V | 66 (33.00) | 26 (30.23) | 40 (35.09) | |
| PESI score (pts) | 113.02±33.84 | 104.59±37.51 | 119.38±29.38 | 0.01 |
| Biochemical and clinical parameters | | | | |
| Heart rate (beats/ minute) | 105.85±22.65 | 106.13±23.22 | 105.63±22.32 | 0.87 |
| Respiratory rate (breaths/ minute) | 26.40±8.19 | 23.67±7.18 | 28.45±8.33 | 0.001 |
| Oxygen saturation (%) | 81.80±14.66 | 87.06±11.31 | 77.83±15.66 | <0.001 |
| High-sensitivity troponin I (pg/mL) | 116.94±295.36 | 104.21±382.25 | 126.54±208.58 | <0.001 |
| Hemoglobin (g/dL) | 12.92±3.29 | 11.99±2.86 | 13.63±3.42 | <0.001 |
| Platelets (10x10³cells) | 239.24±129.33 | 229.72±139.65 | 246.42±121.09 | 0.37 |
| INR | 1.63±3.98 | 1.40±0.75 | 1.79±5.24 | 0.85 |
| Creatinine (mg/dL) | 1.43±2.30 | 1.38±1.74 | 1.47±2.65 | 0.08 |
| Sodium (mmol/dL) | 136.26±5.23 | 136.18±5.99 | 136.30±4.61 | 0.89 |
| Electrocardiographic changes, N (%) | | | | 0.15 |
| Without baseline electrocardiogram | 83 (41.50) | 32 (37.21) | 51 (44.74) | |
| Sinus rhythm | 37 (18.50) | 13 (15.12) | 24 (21.05) | |
| Changes associated with PE | 80 (40.00) | 41 (47.67) | 39 (34.21) | |

*(Continued)*

**Table 1.** (Continued)

| Characteristics | Overall N = 200 | Non COVID-19 season (N = 86) | COVID-19 season (N = 114) | *P* |
|---|---|---|---|---|
| **Other relevant variables, N (%)** | | | | |
| **Stay in Intensive Care** | 82 (41.00) | 31 (36.05) | 51 (44.74) | 0.22 |
| **Mechanic Ventilation** | 45 (22.50) | 10 (11.63) | 35 (30.70) | 0.001 |
| **Hemodynamic Decompensation** | 40 (20.00) | 13 (15.12) | 27 (23.68) | 0.13 |
| **In-hospital death** | 64 (32.00) | 22 (25.58) | 42 (36.84) | 0.09 |
| **Non-adherence** | 79/174 (45.40) | 29/66 (43.94) | 50/108 (46.30) | 0.76 |

Abbreviations: CI, confidence interval; COPD, chronic obstructive pulmonary disease; VTE, venous thromboembolism; DVT, deep vein thrombosis; PESI, Pulmonary Embolism Severity Index; INR, international normalized ratio; PE, pulmonary embolism.

* Plus–minus values are means ±SD.

Notably, when contrasting the RR of adherence in the COVID-19 season vs the non-COVID-19 season, an RR of 1.21 CI95% 0.54–2.91 was obtained, implying that there is no difference between both periods in the incidence of adverse events when definition of adherence was met (Fig 2).

Finally, the population attributable risk of non-adherence for the composite outcome was = 8.30% (95% CI, 1.80–14.73%); for in-hospital death, was = 7.85% (95% CI, 1.50–14.13%), and for hemodynamic decompensation at day 7 was = 5.39% (95% CI, −0.14 to 10.89%) (Fig 3).

## Discussion

To the best of our knowledge, this is the first study to evaluate the guideline adherence of acute pulmonary embolism before, during and shortly after the COVID-19 pandemic-induced hospital-strain. Our results suggest: 1) Guideline adherence deviations were common. 2) The major risk factors for non-adherence were the presence of a higher mortality risk classification and a higher PESI score. 3) Non-adherent management strongly correlated with adverse in-hospital outcomes, especially during the COVID-19 pandemic.

In this study, the presence of global non-adherence was 45.4%. These results differ from those obtained by Jiménez D et al. [8], where they found a global non-adherence of 19%. This difference may be explained by our strict criteria definition of adherence to international recommendations and the greater number of possible adherence deviations we considered. Nevertheless, our results are comparable with those by Bertha Cadena Núñez et al. [15], where they demonstrated that adherence to guidelines was not met in 42% and 86% of patients with acute intermediate-risk and high-risk PE, respectively.

We found that lack of adherence was mostly presented in the next three circumstances: 1) Need for any advanced treatment in patients with high-risk PE with any contraindication to systemic thrombolysis, 2) placement of inferior vena cava filters, and 3) use of any advanced treatment in patients with intermediate-high risk PE with further risk of hemodynamic decompensation and with a low bleeding risk. Although it is well established that patients with high-risk PE and contraindications to systemic fibrinolytic therapy may be considered for reduced dose of systemic thrombolysis, percutaneous catheter-directed treatment (mechanical fragmentation, thrombus aspiration, catheter directed thrombolysis) or surgical embolectomy [3,7,12,16], most of our patients in this scenario did not receive any advanced intervention. Similar findings were demonstrated by Jiménez D et al., where 67% of their high-risk patients did not receive thrombolytic therapy [8]. Currently, the inability to receive anti-coagulation is the most used indication for filter placement [3,7,12,16,17], in our study the 91.43% of patients did not receive a guideline adherent management, a previous study reported that 90% of the time their patients did or did not undergo inferior vena cava filter treatment according to guideline recommendations [8].

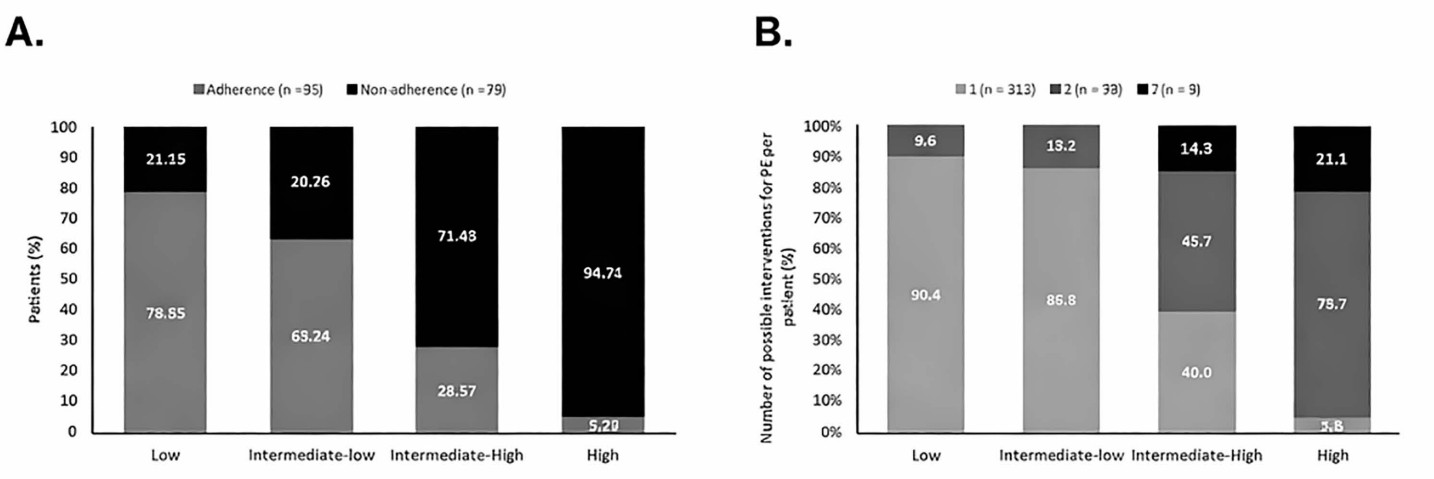

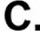

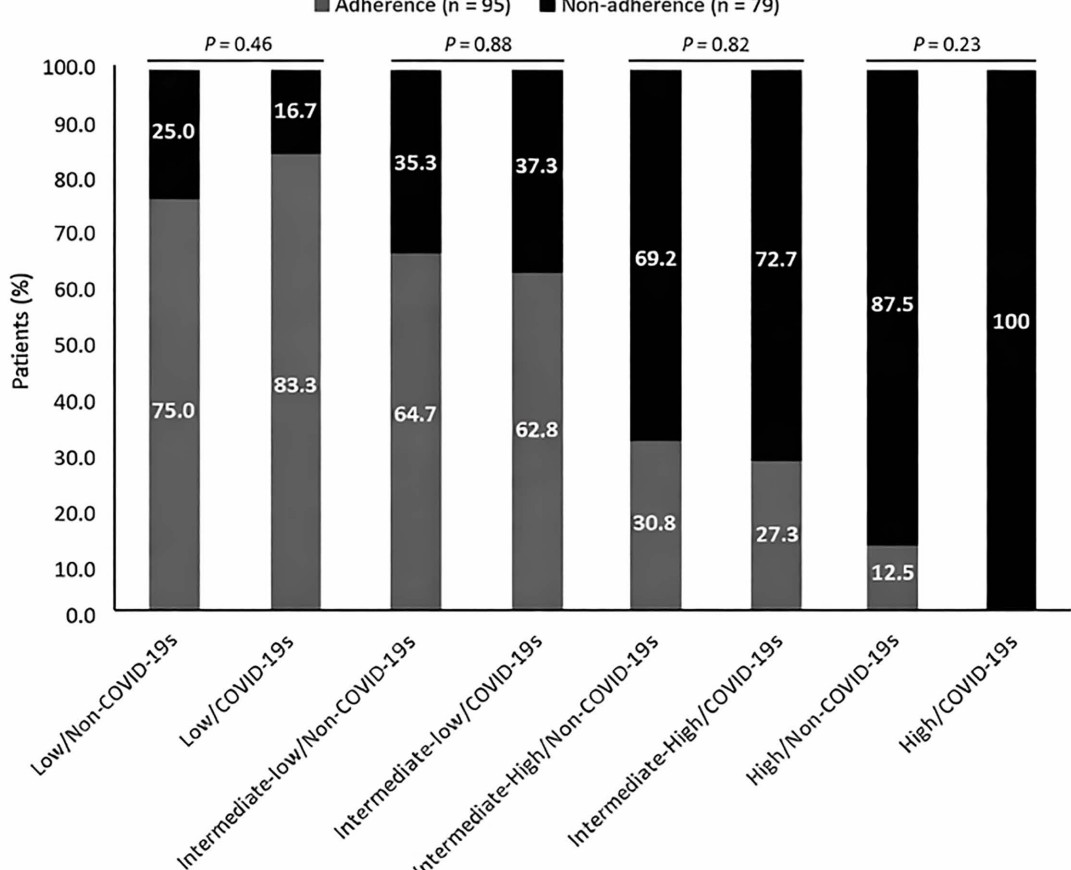

**Fig 1. Stratification according to adherence and mortality risk.** Abbreviations: PE, pulmonary embolism; COVID-19s, COVID-19 season. Panel A, data from the entire study population. Panel B, the potential number of interventions for PE per patient. Panel C, the proportion of individuals is displayed according to adherence, mortality risk, and COVID-19 season.

**Table 2. Incidence rates per 1000 days/person by adherence during the non-COVID-19 and COVID-19 seasons*.**

| | Incidence rate per 1000 days/person (CI, 95%) | | Incidence rate per 1000 days/person (CI, 95%) | |
|---|---|---|---|---|
| | Non – COVID – 19 season (N = 66) | | COVID – 19 season (N = 108) | |
| | Non-adherence (n = 29) | Adherence (n = 37) | Non-adherence (n = 50) | Adherence (n = 58) |
| **Primary outcome** | 21.08 (10.97 - 40.51) | 19.19 (10.33 - 35.67) | 47.47 (33.39 - 67.50) | 23.31 (15.04 - 36.13) |
| **Hemodynamic decompensation** | 11.71 (4.87 - 28.13) | 9.60 (3.99 - 23.06) | 26.03 (16.18 - 41.88) | 11.66 (6.27 - 21.66) |
| **Hemorrhage** | – | 5.76 (1.86 - 17.85) | 1.53 (0.22 - 10.87) | 1.17 (0.16 - 8.27) |
| **Death** | 21.08 (10.97 - 40.51) | 17.27 (8.99 - 33.20) | 39.82 (27.11 - 58.48) | 18.65 (11.42 - 30.44) |
| **PESI class** | | | | |
| I | – | – | – | 17.86 (2.52 - 126.77) |
| II | – | 8.26 (2.07 - 33.04) | – | 24.10 (6.03 - 96.35) |
| III | 20.62 (5.16 - 82.44) | 37.31 (15.53 - 89.65) | 34.97 (14.55 - 84.00) | 36.50 (15.19 - 87.68) |
| IV | 16.39 (2.31 - 116.38) | 50.00 (7.04 - 354.95) | 61.54 (34.95 - 108.36) | 14.88 (6.19 - 35.75) |
| V | 22.99 (10.33 - 51.17) | 33.33 (8.34 - 133.28) | 47.78 (28.30 - 80.68) | 28.46 (13.57 - 59.69) |
| **Mortality risk** | | | | |
| Low | 10.99 (1.55 - 78.01) | 17.92 (7.46 - 43.06) | 47.62 (11.91 - 190.40) | 19.74 (8.87 - 43.93) |
| Intermediate-low | 30.30 (9.77 - 93.96) | 24.54 (9.21 - 65.38) | 49.18 (27.93 - 86.60) | 26.48 (15.37 - 45.60) |
| Intermediate-High | 8.13 (1.15 - 57.72) | 14.08 (1.98 - 99.99) | 59.52 (32.03 - 110.63) | 15.87 (2.24 - 112.68) |
| High | 35.09 (13.17 - 93.49) | – | 35.18 (16.77 - 73.79) | – |

Abbreviations: CI, confidence interval; PESI, Pulmonary Embolism Severity Index.

* If it was not possible to estimate the incidence rate due to the absence of events or person-days, it was indicated with a hyphen (-).

Additionally, there has been an increasing interest in establishing the best management approach for patients with acute PE and hemodynamic stability, as there may be a subgroup of patients that may benefit from advanced treatment strategies, especially those with RV dysfunction and myocardial injury (intermediate-high-risk PE). To date, evidence has shown clinical benefit of full dose systemic thrombolysis in reducing the short-term death from any cause and/or hemodynamic decompensation at the expense of increase bleeding complications [18], with no apparent benefit in mortality, residual symptoms and RV dysfunction at 24 months [19]. This has led to the development of clinical trials trying to close this gap [20,21]. In our study, 88.8% of these cases the patients did not receive any advanced treatment. Overall, this may reflect real-world implementation challenges common to all advanced therapies. However, other barriers to guideline adherence may include lack of knowledge [22], lack of institutional standardization, and the different degree of involvement of the concerning specialists [ 23–26].

The main risks factors for non-adherence were any mortality risk classification above low-risk PE, and a PESI class different above class I with an OR of 3.47 (95% CI 2.07–5.82) and 1.57 (95% CI 1.04–2.37), respectively. This highlights the importance of classifying PE patients according to their risk of short-term mortality [8]. Furthermore, non-adherence was independently associated with worse in-hospital outcomes, especially during the COVID-19 season, with an incidence rate of the composite outcome of 21.08 per 1000 days/person (95% CI 10.97–40.51) and 47.7 per 1000 days/person (95% CI 33.39–67.50).

## Strengths

This is the largest study evaluating the guideline adherence management of patients with acute PE in our country, and the only one with a randomized and balanced strategy that included patients before, during and shortly after the COVID-19 pandemic. Additionally, our criteria for evaluating adherence to international recommendations is stricter than in previous studies.

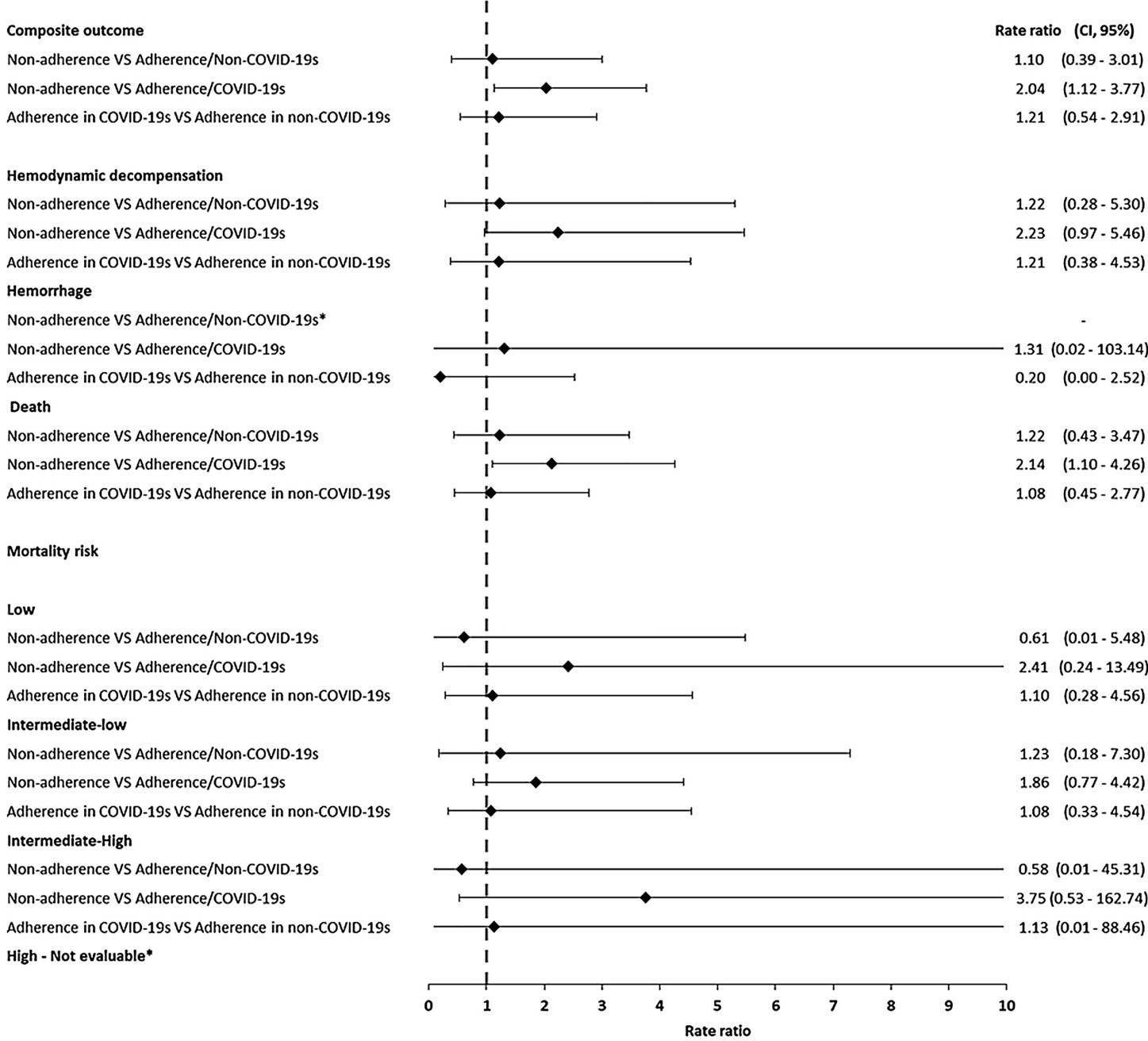

**Fig 2. Rate ratios for the composite and individual outcomes, by mortality risk, and adherence.** Abbreviations: COVID-19s, COVID-19 season.
*= It was not possible to estimate the rate ratio due to the absence of events.

## Limitations

In addition to the intrinsic limitations of any retrospective study, we must acknowledge the following limitations: 1) We conducted this study in an academic center with standardized protocols and continuous medical education programs, which have been shown to impact outcomes. We cannot generalize our findings to non-academic centers. 2) The mortality

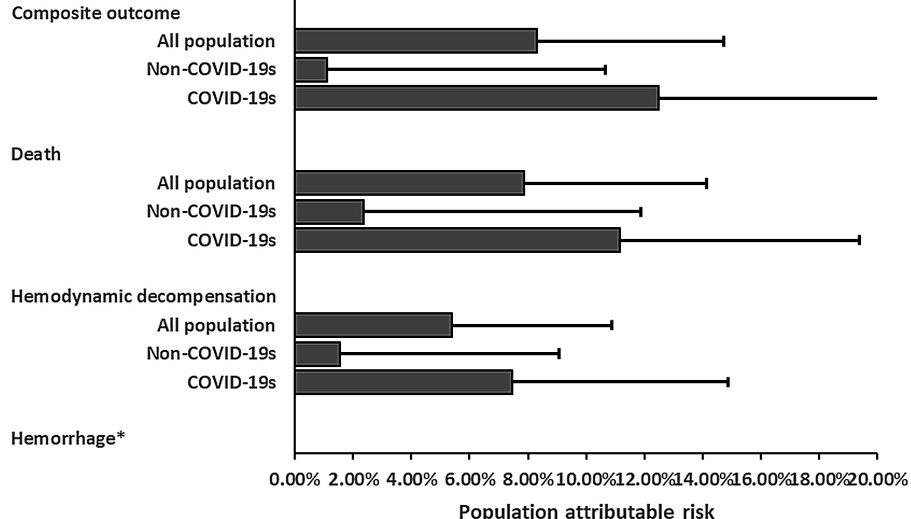

**Fig 3. Population attributable risk for composite and individual outcomes by adherence.** Abbreviations: COVID-19s, COVID-19 season. *= It was not possible to estimate the rate ratio due to the absence of events.

risk and PESI score were calculated by the research team at the time of data collection based on the information reported in the electronic records. 3) We did not collect data to evaluate ICU strain and overflow indicators. 4) We did not assess outcomes beyond hospitalization. 5) We did not consider other indications for parallel treatments (i.e., use of IV UFH infusion as treatment of concomitant acute myocardial ischemia). 6) Before the pandemic, some advanced treatments were not available through our healthcare system, and they had to be paid out-of-pocket, limiting their accessibility during this period. 7) Causality cannot be stated because we did not assess additional potential confounders such as thrombus burden, differences in clinical experience among physicians and among evolving pandemic guidelines. 8) We could not rule-in or out previous existence of right heart failure/pulmonary hypertension during the classification of mortality risk. 9) Our reduced sample size precludes us from making subgroup analyses.

## Conclusions

Guideline adherence deviations were common. The major risk factors for non-adherence were a high mortality risk classification and a higher PESI score, non-adherent management strongly correlated with adverse in-hospital outcomes, especially during the COVID-19 pandemic. Our findings highlight the need of prioritizing the necessary human and material resources to warrant adherence to the standards-of-care of PE patients.

## Supporting information

**S1 Fig. Flowchart of adherence criteria based on international recommendations for the management of acute pulmonary embolism.**
(DOCX)

**S1 Table. Overall adherence to international recommendations by category.**
(DOCX)

**S2 Table. Multivariate model for the composite outcome.**
(DOCX)

## Author contributions

**Conceptualization:** Adrian Soto-Mota.

**Data curation:** Juan José Rodríguez-Crespo, Eduardo Gutiérrez-León, Pedro Dammann-Beltrán, Jurhiat Alejandro Seaman-Gómez, Sergio Contreras-Garduño, Erick Yasar Zúñiga-González, Fausto Alfredo Ríos Olais, Adrian Soto-Mota.

**Formal analysis:** Juan José Rodríguez-Crespo, Eduardo Gutiérrez-León, Jurhiat Alejandro Seaman-Gómez, José Guillermo Domínguez-Cherit, Adrian Soto-Mota.

**Investigation:** Juan José Rodríguez-Crespo, Eduardo Gutiérrez-León, Pedro Dammann-Beltrán, Jurhiat Alejandro Seaman-Gómez, Sergio Contreras-Garduño, Erick Yasar Zúñiga-González, Thierry Hernández-Gilsoul, José de Jesús Vidal-Mayo, Fausto Alfredo Ríos Olais, Raúl Rivera Moscoso, Jorge Oseguera Moguel, Mónica Chapa Ibargüengoitia, Heber Rodríguez Bautista, José Sifuentes-Osornio, Adrian Soto-Mota.

**Methodology:** Juan José Rodríguez-Crespo, Adrian Soto-Mota.

**Project administration:** Juan José Rodríguez-Crespo, Adrian Soto-Mota.

**Resources:** José Sifuentes-Osornio.

**Software:** Adrian Soto-Mota.

**Supervision:** José Guillermo Domínguez-Cherit, Thierry Hernández-Gilsoul, José de Jesús Vidal-Mayo, Raúl Rivera Moscoso, Jorge Oseguera Moguel, Mónica Chapa Ibargüengoitia, Heber Rodríguez Bautista, José Sifuentes-Osornio, Adrian Soto-Mota.

**Validation:** Adrian Soto-Mota.

**Visualization:** Juan José Rodríguez-Crespo, Adrian Soto-Mota.

**Writing – original draft:** Juan José Rodríguez-Crespo, Adrian Soto-Mota.

**Writing – review & editing:** Juan José Rodríguez-Crespo, Eduardo Gutiérrez-León, Pedro Dammann-Beltrán, Jurhiat Alejandro Seaman-Gómez, Sergio Contreras-Garduño, Erick Yasar Zúñiga-González, José Guillermo Domínguez-Cherit, Thierry Hernández-Gilsoul, José de Jesús Vidal-Mayo, Fausto Alfredo Ríos Olais, Raúl Rivera Moscoso, Jorge Oseguera Moguel, Mónica Chapa Ibargüengoitia, Heber Rodríguez Bautista, José Sifuentes-Osornio, Adrian Soto-Mota.

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
