## [Decision Letter · Decision Letter 0]

29 Dec 2025

Dear Dr. Soto-Mota,

We look forward to receiving your revised manuscript.

Kind regards,

Eyüp Serhat Çalık

Academic Editor

PLOS One

Journal Requirements:

7.Please review your reference list to ensure that it is complete and correct. If you have cited papers that have been retracted, please include the rationale for doing so in the manuscript text, or remove these references and replace them with relevant current references. Any changes to the reference list should be mentioned in the rebuttal letter that accompanies your revised manuscript. If you need to cite a retracted article, indicate the article’s retracted status in the References list and also include a citation and full reference for the retraction notice.

Additional Editor Comments :

Dear Authors,

I commend your research on the relationship between the burden of patients treated for pulmonary embolism during the Covid-19 pandemic and compliance with international guidelines. Your manuscript has been reviewed by two external reviewers, and their comments are provided below. Please provide point-by-point responses to the comments and make the appropriate revisions to your manuscript accordingly. We look forward to receiving your revised manuscript. Best of luck.

Reviewers' comments:

Reviewer's Responses to Questions

**Comments to the Author**

1. Is the manuscript technically sound, and do the data support the conclusions?

Reviewer #1: Partly

Reviewer #2: Yes

2. Has the statistical analysis been performed appropriately and rigorously?

Reviewer #1: Yes

Reviewer #2: Yes

3. Have the authors made all data underlying the findings in their manuscript fully available?

Reviewer #1: Yes

Reviewer #2: Yes

4. Is the manuscript presented in an intelligible fashion and written in standard English?

Reviewer #1: Yes

Reviewer #2: Yes

Reviewer #1: The study is good and is extended over three time periods.It allows meaningful comparison of adherence to the guidelines across time zones- during periods of strain and during routine care. However, the sample size is small and hence subgroup analysis is not possible. Also, important measures of echo like pulmonary artery pressures, right ventricular dysfunction is not mentioned and I would like to know how the risk stratification was done?The study being retrospective , many cinfounders like clinical experience of the physicians , evolving pandemic guidelines were not taken into account.

Because of the nature of the study design, causality between non adherence to the guidelines and outcomes cannot be established.

Reviewer #2: the authors have done the study with the aim too see the outcome of Pulmonary embolism in COvid 19 and relation with guidlines the .study done with proper research tools applies ,written and done professionally.statistical tool applied scientifically and properly .

conclusion is in line with objective of the study and the statistical too applied with also in line with objective of the study and the methodology use,.results are explained with proper research and statisticwording .discsuiojn written and explained as profesionla write up references quoted right and with porper litersture search .

thus study it will also be of benefit to health professional ,working with in emergency with PE and COVID 19 or in f influenza outbreak

.

Reviewer #1: No

Reviewer #2: No

---

## [Author Response · Author response to Decision Letter 1]

18 Feb 2026

Eyüp Serhat Çalık

Academic Editor

PLOS One

We sincerely appreciate the opportunity to revise and improve our manuscript entitled “Outcomes in acute pulmonary embolism and their association with adherence to international recommendations around COVID-19 pandemic-induced hospital-strain: The experience in a Mexican tertiary care center.” and for the time and effort invested by you and the reviewers in evaluating our work.

We have carefully reviewed all your comments and those of the reviewers. Below, we provide a response to each comment. To facilitate their review, all editorial-team comments are displayed in red. Our manuscript has been revised accordingly, and all changes made in the enclosed manuscript are indicated in highlighted text in the revised version. To facilitate reviewing these changes, we added the corresponding page numbers in our responses, too.

We hope this current version of our manuscript now meets the standards to be eligible for publication in your journal. Nonetheless, we would be happy to provide any additional information if required. Thank you once again for the time, effort, and consideration granted to our submission.

Dr. Adrian Soto-Mota, MD, PhD, FACP.

Chief of the Metabolic Diseases Research Unit

National Institute of Medical Science and Nutrition Salvador Zubirán

adrian.sotom@incmnsz.mx,

phone: +5255-5487-0900, ext: 6319

Reviewer #1: The study is good and is extended over three time periods.It allows meaningful comparison of adherence to the guidelines across time zones- during periods of strain and during routine care. However, the sample size is small and hence subgroup analysis is not possible.

Thank you for raising this point. We included this limitation in the relevant section (page 15, line 332).

Also, important measures of echo like pulmonary artery pressures, right ventricular dysfunction is not mentioned and I would like to know how the risk stratification was done?

Thank you for raising this point. We totally agree with the reviewer on the importance of echo-derived pulmonary artery pressures for risk stratification. However, these values were not always available in the eligible medical records. As mentioned in our Methods section (page 6, line 146), risk stratification was done with PESI, which was calculated by the research team (at data collection). We acknowledge this limitation in the relevant section (page 15, lines 321-323).

The study being retrospective , many cinfounders like clinical experience of the physicians , evolving pandemic guidelines were not taken into account. Because of the nature of the study design, causality between non adherence to the guidelines and outcomes cannot be established.

Thank you for raising this point. We included this limitations in the relevant section (page 15, line 328-330).

Reviewer #2: the authors have done the study with the aim too see the outcome of Pulmonary embolism in COvid 19 and relation with guidlines the .study done with proper research tools applies ,written and done professionally.statistical tool applied scientifically and properly .

conclusion is in line with objective of the study and the statistical too applied with also in line with objective of the study and the methodology use,.results are explained with proper research and statisticwording .discsuiojn written and explained as profesionla write up references quoted right and with porper litersture search .

thus study it will also be of benefit to health professional ,working with in emergency with PE and COVID 19 or in f influenza outbreak

Thank you for your positive feedback. We also think these insights could prove beneficial for other infectious outbreaks beyond COVID-19, such as influenza.

---

## [Decision Letter · Decision Letter 1]

8 Apr 2026

Outcomes in acute pulmonary embolism and their association with adherence to international recommendations around COVID-19 pandemic-induced hospital-strain: The experience in a Mexican tertiary care center.

PONE-D-25-57661R1

Dear Dr. Soto-Mota,

We’re pleased to inform you that your manuscript has been judged scientifically suitable for publication and will be formally accepted for publication once it meets all outstanding technical requirements.

Kind regards,

Eyüp Serhat Çalık

Academic Editor

PLOS One

Additional Editor Comments (optional):

Reviewers' comments:

Reviewer's Responses to Questions

**Comments to the Author**

Reviewer #1: All comments have been addressed

Reviewer #2: All comments have been addressed

2. Is the manuscript technically sound, and do the data support the conclusions?

Reviewer #1: Yes

Reviewer #2: Yes

3. Has the statistical analysis been performed appropriately and rigorously?

Reviewer #1: Yes

Reviewer #2: Yes

4. Have the authors made all data underlying the findings in their manuscript fully available?

Reviewer #1: Yes

Reviewer #2: Yes

5. Is the manuscript presented in an intelligible fashion and written in standard English?

Reviewer #1: Yes

Reviewer #2: No

Reviewer #1: All comments have been addressed to satisfaction and the manuscript may be accepted if found suitable by the other reviewers and the editorial board.

Reviewer #2: All the comments have been addressed according to the questions rais

ed .introduction is explained. Manuscript has been corrected according to the comments.results are explained and statistics applied properly. Data is provided with manuscript.

Its written int

.

Reviewer #1: No

Reviewer #2: No

---

## [Editor Report · Acceptance letter]

PONE-D-25-57661R1

PLOS One

Dear Dr. Soto-Mota,

I'm pleased to inform you that your manuscript has been deemed suitable for publication in PLOS One. Congratulations! Your manuscript is now being handed over to our production team.

Kind regards,

on behalf of

Dr. Eyüp Serhat Çalık

Academic Editor

PLOS One